# Targeting Cancer Metabolism Breaks Radioresistance by Impairing the Stress Response

**DOI:** 10.3390/cancers13153762

**Published:** 2021-07-27

**Authors:** Melissa Schwab, Katharina Thunborg, Omid Azimzadeh, Christine von Toerne, Caroline Werner, Maxim Shevtsov, Tommaso Di Genio, Masa Zdralevic, Jacques Pouyssegur, Kathrin Renner, Marina Kreutz, Gabriele Multhoff

**Affiliations:** 1Radiation Immuno-Oncology Group, Center for Translational Cancer Research (TranslaTUM), School of Medicine, Klinikum rechts der Isar, Technical University of Munich (TUM), 81675 Munich, Germany; melissa.schwab@tum.de (M.S.); katharina.thunborg@tum.de (K.T.); c.werner@tum.de (C.W.); Maxim.shevtsov@tum.de (M.S.); tommaso.genio@tum.de (T.D.G.); 2German Research Center for Environmental Health, Institute of Radiation Biology, Helmholtz Zentrum München, 85764 Neuherberg, Germany; oazimzadeh@bfs.de; 3Section Radiation Biology, Federal Office for Radiation Protection (BfS), 85764 Neuherberg, Germany; 4Research Unit Protein Science, Helmholtz Center Munich, German Research Center for Environmental Health, 85764 Neuherberg, Germany; vontoerne@helmholtz-muenchen.de; 5Institute of Cytology, Institute of Russian Academy of Sciences (RAS), 194064 St. Petersburg, Russia; 6Department of Biotechnology, Pavlov First Saint Petersburg State Medical University, 197022 St. Petersburg, Russia; 7Faculty of Medicine, University of Montenegro, Kruševac, 81000 Podgorica, Montenegro; Masa.Zdralevic@unice.fr; 8Institute for Research on Cancer and Aging, University Côte d’Azur, CNRS, INSERM, Centre Antoine Lacassagne, 06107 Nice, France; jacques.pouyssegur@unice.fr; 9Department of Medical Biology, Centre Scientifique de Monaco (CSM), 98000 Monaco, Monaco; 10Department of Internal Medicine III, University of Regensburg, 93053 Regensburg, Germany; kathrin.renner@klinik.uni-regensburg.de (K.R.); Marina.Kreutz@klinik.uni-regensburg.de (M.K.); 11Center for Interventional Immunology, Department of Internal Medicine III, University of Regensburg (RCI), 93053 Regensburg, Germany; 12Department of Radiation Oncology, School of Medicine, Klinikum rechts der Isar, Technical University of Munich (TUM), 81675 Munich, Germany

**Keywords:** *LDHA/B* double knockout, lactate/pyruvate metabolism, radiosensitivity, stress response, membrane heat shock protein 70 (Hsp70)

## Abstract

**Simple Summary:**

Ionizing radiation is a major pillar in the therapy of solid tumors. However, normal tissue toxicities and radioresistance of tumor cells can limit the therapeutic success of radiotherapy. In this study, we investigated the coregulation of the cancer metabolism and the heat shock response with respect to radioresistance. Our results indicate that an inhibition of lactate dehydrogenase, either pharmacologically or by gene knockout of *LDHA* and *LDHB*, significantly increases the radiosensitivity in tumor cells by global impairing of the stress response. Therefore, inhibition of the lactate metabolism might provide a promising strategy in the future to improve the clinical outcome of patients with highly aggressive, therapy-resistant tumors.

**Abstract:**

The heightened energetic demand increases lactate dehydrogenase (LDH) activity, the corresponding oncometabolite lactate, expression of heat shock proteins (HSPs) and thereby promotes therapy resistance in many malignant tumor cell types. Therefore, we assessed the coregulation of LDH and the heat shock response with respect to radiation resistance in different tumor cells (B16F10 murine melanoma and LS174T human colorectal adenocarcinoma). The inhibition of LDH activity by oxamate or GNE-140, glucose deprivation and *LDHA/B* double knockout (LDH^−^^/^^−^) in B16F10 and LS174T cells significantly diminish tumor growth; ROS production and the cytosolic expression of different HSPs, including Hsp90, Hsp70 and Hsp27 concomitant with a reduction of heat shock factor 1 (HSF1)/pHSF1. An altered lipid metabolism mediated by a *LDHA/B* double knockout results in a decreased presence of the Hsp70-anchoring glycosphingolipid Gb3 on the cell surface of tumor cells, which, in turn, reduces the membrane Hsp70 density and increases the extracellular Hsp70 levels. Vice versa, elevated extracellular lactate/pyruvate concentrations increase the membrane Hsp70 expression in wildtype tumor cells. Functionally, an inhibition of LDH causes a generalized reduction of cytosolic and membrane-bound HSPs in tumor cells and significantly increases the radiosensitivity, which is associated with a G2/M arrest. We demonstrate that targeting of the lactate/pyruvate metabolism breaks the radioresistance by impairing the stress response.

## 1. Introduction

Due to their fast growth rates and high energetic demand [1], the uptake of glucose and glycolytic capacity in tumor cells (Warburg effect) is greater than that in normal cells [2]. An increased conversion of glucose to pyruvate and the formation of high lactate concentrations caused by an enhanced lactate dehydrogenase A (LDHA) activity leads to an acidic tumor microenvironment that promotes an aggressive tumor phenotype and an increased risk for metastases and tumor recurrence [3,4]. In addition to pyruvate and lactate, a KRAS-mediated altered glutamine metabolism has been found to cause chemoresistance in pancreatic cancers [5]. Therefore, targeting the glutamine metabolism [5], as well as silencing the activity of LDHA, provide promising strategies to reduce tumor progression, as demonstrated in preclinical tumor models [6]. In addition to LDHA, the activity of lactate dehydrogenase B (LDHB), which also supports the conversion of lactate to pyruvate, is frequently upregulated in tumor cells [7,8] and can thereby mediate therapy resistance [9]. Moreover, the expression of HSPs in general, but especially that of the major stress-inducible Hsp70, is frequently increased in a large variety of different tumor cell types, in which they fulfill chaperoning functions that contribute to tumor cell survival and protection against the lethal damage of environmental stress [10,11,12,13]. However, despite its potential importance, insight into the relationship between the pro-tumorigenic lactate metabolism and the antiapoptotic stress response is limited. In immature boar Sertoli cells, a heat-induced upregulation of Hsp70, glucose transporter 3 (Glut-3) and lactate production has been shown to improve spermatogenesis and male fertility via LDHA [14], and high cytosolic Hsp70 levels in HeLa human cervical cancer cells shift the energy metabolism towards anaerobic glycolysis [15]. The mechanistic link between the heat shock response, lactate metabolism and sensitivity of tumor cells to radiation has not yet been elucidated.

Apart from surgery and chemo- and immunotherapy, ionizing radiation remains one of the major therapeutic pillars for solid tumors [16]. However, normal tissue toxicities and the radioresistance of tumor cells hamper a favorable clinical outcome for radiotherapy in many clinical settings. In addition to hypoxia-induced transcriptional responses, such as the activation of hypoxia-inducible factor alpha (HIF-1a) and PI3K/Akt/mTOR pathways [17], high lactate concentrations [18] and a radiation-induced upregulation of the heat shock response, including the major stress-inducible, antiapoptotic protein Hsp70, are known factors that contribute to radiation resistance [19].

## 2. Materials and Methods

### 2.1. Cells and Cell Culture

The mouse B16F10 wildtype melanoma cell line (ATCC^®^ CRL-6475™; ATCC, Manassas, VA, USA) (B16 WT, 0.008 × 10^6^ cells/mL) and *LDHA/B* double knockout cell line (B16 LDH^−/−^, 0.012 × 10^6^ cells/mL) and the human wildtype LS174T colorectal adenocarcinoma cell line (ATCC^®^ CL-188™; ATCC, Manassas, VA, USA) (LS174T WT, 0.06 × 10^6^ cells/mL) and *LDHA/B* double knockout cell line (LS174T LDH^−/−^, 0.12 × 10^6^ cells/mL)—kindly provided by Marina Kreutz, Jacques Pouyssegur [20]—were cultured in Roswell Park Memorial Institute (RPMI)-1640 Medium (Sigma-Aldrich, St. Louis, MO, USA) and high-glucose Dulbecco’s Eagle’s Minimum Essential Medium (DMEM) (Sigma-Aldrich) supplemented with 10% *v/v* heat-inactivated fetal bovine serum (FBS) (Sigma-Aldrich), 1% antibiotics (10,000-IU/mL penicillin and 10-mg/mL streptomycin, Sigma-Aldrich), 2-mM L-glutamine (Sigma-Aldrich) and 1-mM sodium pyruvate (Sigma-Aldrich). Cells were routinely checked for mycoplasma contamination.

### 2.2. Reagents and Treatment

The pyruvate analog inhibitor of gluconeogenesis and glycolysis, sodium oxamate (Oxa) (Santa Cruz, Dallas, TX, USA), was dissolved in the relevant cell culture medium in which the cells were grown, and a 4-mM stock solution of the novel LDHA/B/C inhibitor GNE-140 (GNE, LDHA/B/C IC50 = 3/5/5 nM) (Sigma-Aldrich) was prepared in DMSO (Sigma-Aldrich). Tumor cells were incubated with a sublethal concentration of Oxa (60 mM) for 48 h or GNE (10 µM) for 24 h. A stock solution (1 M) of sodium lactate (NaLac) (Sigma-Aldrich) was prepared in H_2_O, and a 100-mM sodium pyruvate solution was purchased from Sigma-Aldrich. Cells were incubated with 15-mM NaLac or pyruvate for 6 h. Control cells were incubated with the respective amounts of diluents, media or DMSO, as appropriate.

### 2.3. Lactate Dehydrogenase (LDH) Activity Measurement

LDH activity was determined using the Lactate Dehydrogenase Activity kit (Sigma-Aldrich) following the manufacturer’s protocols.

### 2.4. Western Blot Analysis

Cells were lysed in a radioimmunoprecipitation assay (RIPA) buffer containing 50-mM Tris-HCl (pH 8.0), 150-mM NaCl, 1-mM EDTA, 1% *v*/*v* Triton-X-100, 0.1% *w*/*v* sodium dodecyl sulphate (SDS), 0.5% *w*/*v* sodium deoxycholate and protease inhibitor cocktail (Roche, Basel, Switzerland). The protein content was determined using the Pierce™ BCA Protein Assay Kit (Thermo Fisher Scientific, Waltham, MA, USA). Proteins were separated on a SDS-PAGE, transferred onto nitrocellulose membranes and transferred proteins detected by immunoblotting using the following primary and secondary antibodies: anti-HSF1 (ADI-SPA-901-D; Enzo Life Sciences, Farmingdale, NY, USA), anti-HSF1 phospho-S326 (ab76076, Abcam, Cambridge, MA, USA), anti-Hsp27 (NBP2-32972, Novus Biologicals, Centennial, CO, USA), anti-Hsp70 (cmHsp70.1, IgG1, multimmune GmbH, Munich, Germany), anti-Hsp90 (4874, Cell Signaling Technology, Danvers, MA, USA), anti-LDHA (NBP1-48336, Novus Biologicals, Centennial, CO, USA), anti-LDHB (NBP2-53421, Novus Biologicals), anti-β-Actin (A2228, Sigma-Aldrich), HRP-conjugated rabbit anti-mouse immunoglobulins (P0260, Dako-Agilent, Santa Clara, CA, USA) and HRP-conjugated swine anti-rabbit immunoglobulins (P0217, Dako-Agilent, Santa Clara, CA, USA). Immune complexes were detected by a Pierce™ ECL Western Kit (Thermo Fisher Scientific) and imaged digitally (ChemiDoc™ Touch Imaging System, Bio-Rad, Hercules, CA, USA). Fiji software (https://imagej.net/software/fiji/, accessed on 22 April 2021) [21] was used for quantification of the Western blot signals.

### 2.5. Flow Cytometry

Expression of membrane Hsp70 (mHsp70) was determined by flow cytometry using the FITC-conjugated cmHsp70.1 monoclonal antibody (mAb, IgG1, multimmune GmbH, Munich, Germany) on a FACSCalibur™ flow cytometer (BD Biosciences, Heidelberg, Germany). Briefly, after washing with flow cytometry buffer (phosphate-buffered saline (PBS) containing 10% *v*/*v* FBS), viable tumor cells (0.2 × 10^6^ cells) were incubated either with the cmHsp70.1 mAb or an isotype matched FITC-labeled control antibody (mouse IgG1 FITC, 345815; BD Biosciences) on ice in the dark for 30 min. Only viable (propidium iodide (PI)-negative) cells with intact cell membranes were gated upon and analyzed.

For analyzing the expression of the lactate transporter MCT1 by flow cytometry, LS174T cells were permeabilized with methanol and stained using the APC-conjugated human MCT1/SLC16A1 monoclonal antibody (mAb, IgG2A; R&D Systems, Minneapolis, MN, USA) or an isotype matched APC-labeled control antibody (mouse IgG2A; Beckman Coulter, Brea, CA, USA).

### 2.6. Dichlorodihydrofluorescein Diacetate (DCFDA) Assay for Measuring Reactive Oxygen Species

The DCFDA Cellular ROS Detection Assay Kit (Abcam) was used to measure the intracellular levels of reactive oxygen species (ROS). Fluorescence signals were determined on a VICTOR Multilabel plate reader (PerkinElmer, Waltham, MA, USA) with 485/535-nm excitation/emission filters. The average relative fluorescence signal of WT cells was set to 1 and that of LDH^−/−^ cells calculated proportionally.

### 2.7. Cell Proliferation Assay

Cell proliferation was measured using a Sigma-Aldrich Cell Counting Kit-8 (CCK-8), according to the manufacturer’s protocols.

### 2.8. Immunocytochemistry

LS174T cells were grown on poly-L-lysine-coated glass slides. After blocking with 5% *w/v* bovine serum albumin (BSA) in PBS, cells were co-stained with FITC-labeled cmHsp70.1 mAb (multimmune GmbH) and PE-labeled CD77 (Gb3) mAb (563631, BD Biosciences) on ice for 20 min. Cells were washed with ice-cold PBS and fixed in 0.5% *w/v* paraformaldehyde in PBS. Nuclei were counter-stained with Hoechst 33342 (H3570, Invitrogen, Carlsbad, CA, USA). Fluorescence images were taken using a Leica TCS SP8 confocal microscope.

### 2.9. Measurement of Extracellular Hsp70 Levels

Extracellular Hsp70 levels in the supernatant of LS174T cells were determined by ELISA (R&D Systems) following the manufacturer’s recommendations. The measured values of extracellular Hsp70 were normalized to 1 × 10^6^ viable tumor cells.

### 2.10. Proteomics

Proteins were digested using the filter-aided sample preparation (FASP) protocol [22,23] and digested peptides analyzed by LC-MS/MS in the data-dependent acquisition (DDA) mode. MS data were acquired on a Q-Exactive HF-X mass spectrometer (Thermo Fisher Scientific) coupled to an Ultimate 3000 nano-RSLC (Thermo Fisher Scientific; Dionex™, Waltham, MA, USA) [24]. Proteome Discoverer (PD) 2.4 software (Thermo Fisher Scientific; version 2.4.1.15, Thermo Fisher Scientific, Waltham, MA, USA) was used for peptide and protein identification via a database search (Sequest HT search engine; distributed by Thermo Fisher Scientific, Waltham, MA, USA) against the Swiss-Prot human database (Release 2020_02, 20349 sequences in PD). The peptide spectrum matches and peptides were performed by accepting only the top-scoring hit for each spectrum and satisfying the cut-off values for FDR <1% and a posterior error probability <0.01. The final protein ratio was calculated using the median abundance values of 4 replicates for each of the experimental groups (LS174T LDH^−/−^ versus WT). The statistical significance of the ratio change was ascertained by employing the *t*-test approach, as described previously [25]. The proteins identified and quantified with at least 2 unique peptides and with ratios greater than 1.30-fold or less than 0.77-fold (*t*-test; *p* < 0.05) were defined as being significantly differentially expressed for the final quantification. Details on the protein preparation, identification and quantification are provided in the Appendix A section.

### 2.11. Interaction and Signaling Network Analysis

Protein–protein interaction and signaling networks analyses were performed using Ingenuity Pathway Analysis (IPA) software (QIAGEN Inc., Hilden, Germany; https://www.qiagenbioinformatics.com/products/ingenuity-pathway-analysis, accessed on 22 April 2021) [26].

### 2.12. Irradiation

Tumor cells were irradiated with a single dose of 0 Gy (sham), 2 Gy, 4 Gy and 6 Gy using the CellRad compact benchtop X-ray irradiator (Precision X-Ray, North Branford, CT, USA) at a dose rate of 1 Gy/min (5 mA, 130 kV).

### 2.13. Analysis of Cell Survival Using the Clonogenic Assay

Tumor cells were seeded into 12-well plates and irradiated with the indicated doses 24 h after seeding (48 h for LDH inhibitor-treated cells). After treatment with the LDH inhibitor and irradiation, cells were washed and incubated in drug-free medium. After 5–9 days, plates were washed with PBS, fixed with ice-cold methanol and colonies were stained with 0.1% *w/v* crystal violet. The number of colonies (≥50 cells) was measured automatically in a Bioreader^®^ 3000 (Bio-Sys GmbH, Karben, Germany). Survival curves were fitted to the linear quadratic model using SigmaPlot (Systat Software Inc, San Jose, CA, USA).

### 2.14. Cell Cycle Analysis

For the cell cycle analysis, cells were fixed in 70% *v/v* methanol overnight at 4 °C, incubated with RNase for 15 min at 37 °C, stained with propidium iodide (PI) and analyzed using a FACSCalibur™ flow cytometer (BD Biosciences).

### 2.15. Statistics

Each experiment was independently performed at least 3 times (biological replicates). The Student’s *t*-test was used to evaluate significant differences between two groups. When comparing multiple groups, Tukey’s test was applied (*: *p* ≤ 0.05, **: *p* ≤ 0.01 and ***: *p* ≤ 0.001). Data are presented as mean values with standard deviations (SD).

## 3. Results

### 3.1. LDH Inhibitors, Glucose Deprivation and a LDHA/B Double Knockout Significantly Reduce Cytosolic HSPs

The LDH activity in LS174T WT tumor cells was inhibited with sublethal doses of the pyruvate analog Oxa (60 mM, 48 h) and the novel LDHA/B/C inhibitor GNE [27] (10 µM, 24 h) (Figure 1a). The protein levels of the stress proteins Hsp90, Hsp70 and Hsp27 were concomitantly downregulated upon LDH inhibition with both inhibitors (Figure 1b). These data are supported by the findings of Manerba et al., which reported a decrease in HSPs by Oxa in hepatocellular carcinoma cells [28].

To confirm that the effects of Oxa or GNE on Hsp90, Hsp70 and Hsp27 are strictly mediated by LDH inhibition, the LDHA and LDHB double knockout tumor cell lines (murine B16F10 and human LD174T) were used for further studies. In addition to the absence of the enzymes LDHA and LDHB (LDH^−/−^) (Figure 2a) [20], the LDH activity was significantly reduced in both LDH^−/−^ tumor cell lines (Figure 2b), although the expression of the lactate transporter MCT1 remained unaffected by the CRISPR/Cas9-induced LDHA/B double knockout (Figure 2c).

Similar to the results obtained with LDH inhibitors, a *LDHA/B* double knockout reduced the cytosolic Hsp90, Hsp70 and Hsp27 levels in B16F10 and LS174T cells, although the Hsp27 levels could not be detected in B16F10 cells due to a very low Hsp27 expression in murine tumor cells (Figure 3a,b). Consistent with these findings, the expression of non-phosphorylated and phosphorylated Heat Shock Factor 1 (HSF1 and pHSF1, respectively) known to regulate the expression of Hsp70 and Hsp27 was also reduced by a *LDHA/B* double knockout in both tumor cell types (Appendix A).

The connection between metabolic activity, ROS production [29] and HSP expression was demonstrated by a significantly increased proliferative capacity and ROS production in LS174T WT cells with high HSP levels compared to LS174T LDH^−/−^ cells (Figure 3c,d). Consistent with these results, LS174T WT cells cultured in high- (4500 mg/L) versus low-glucose (1000 mg/L) cell culture medium exhibited a significantly higher proliferative capacity (Appendix A); an increased ROS production (Appendix A) and higher Hsp90, Hsp70 and Hsp27 levels (Appendix A).

### 3.2. LDH Inhibitors, Glucose Deprivation and a LDHA/B Double Knockout Reduce Plasma Membrane-Bound Hsp70 and the Glycosphingolipid Gb3

Tumor cells, in contrast to normal cells, not only overexpress Hsp70 in the cytosol but also present it on their plasma membranes [30,31] and actively release it in exosomes [19]. We demonstrated that this tumor-specific membrane localization of Hsp70 is enabled by a tumor-specific lipid composition [32]. Apart from a significant reduction in cytosolic Hsp70 levels, the flow cytometry analysis revealed a significantly reduced membrane expression of Hsp70 upon the treatment of LS174T cells with nonlethal concentrations of the LDH inhibitors Oxa and GNE (Figure 4a). Identical results were observed in LS174T WT cells cultured in a low-glucose medium (Appendix A) and in B16F10 LDH^−/−^ and LS174T LDH^−/−^ tumor cells (Figure 4b). Vice versa, a treatment of tumor cells with an excess of sodium lactate (NaLac, 15 mM) or pyruvate (15 mM) for 6 h significantly increased the membrane Hsp70 expression in LS174T WT but not in LDH^−/−^ cells (Appendix A), whilst cytosolic Hsp70 levels remained unaltered. This finding suggests that the membrane Hsp70 expression might be affected by the LDHA/B activity. A comparative global profiling of the membrane lipids in tumor versus normal cells and artificial lipid cupellation assays revealed that the membrane localization of Hsp70 in tumor cells is enabled by the tumor-specific glycosphingolipid globotriaosylceramide (Gb3/CD77) that resides in cholesterol-rich microdomains, also termed lipid rafts [32]. A potential link between lactate metabolism and membrane Hsp70/Gb3 expression was explored in LS174T WT and LDH^−/−^ tumor cells by confocal microscopy using cmHsp70.1-FITC (green) and CD77-PE (red) mAbs, respectively. As illustrated in Figure 4c, a decreased membrane Hsp70 density was associated with a dramatic decrease in the expression of Gb3 on the membrane of LS174T LDH^−/−^cells. A merge of green Hsp70 and red Gb3 staining revealed a colocalization of both molecules (yellow) on the plasma membrane of LS174T WT cells (Figure 4c). The dotted staining pattern, which is typical for lipid rafts, suggests that both molecules reside in cholesterol-rich microdomains. Increased Hsp70 levels in the extracellular milieu of LS174T LDH^−/−^ cells compared to LS174T WT cells (Figure 4d) might be explained by a lower capacity of Gb3 to anchor Hsp70 in the plasma membrane.

To elucidate the potential mechanisms responsible for the reduced Gb3 levels in the membrane of LDH^−/−^ cells, the cellular proteomes of LS174T WT and LDH^−/−^ knockout cells were comparatively analyzed using label-free quantitative proteomics. In total, 3463 (FDR 5%) proteins were identified and quantified, from which 78% were quantified with at least two unique peptides across multiple samples (Appendix A). Among the quantified proteins, 414 proteins were significantly differentially expressed in LDH^−/−^ versus WT cells (+1.3-fold and *p*-value < 0.05), with the expression of 204 proteins being decreased and that of 210 proteins increased (Appendix A). A detailed analysis of the functional interactions and biological pathways performed with IPA software revealed major differences in the metabolic, lipidomic and stress-related biomolecules between LS174T WT and LDH^−/−^ cells. The most affected pathways in LDH^−/−^ cells are associated with cholesterol biosynthesis; mitochondrial dysfunction and lipid-, carbohydrate- nucleic acid- and protein-metabolism, followed by NRF2-mediated oxidative stress, protein ubiquitination and PPARα-mediated gene regulation (Figure 4e and Appendix A). Another important group of deregulated proteins are associated with the lipid and carbohydrate metabolism, including lipid and cholesterol synthesis, lipid transport, lipid oxidation/reduction and the release of glycosphingolipids (Appendix A). These latter findings might explain the differences in the Gb3 expression in LS174T WT and LDH^−/−^ cells as visualized by confocal fluorescence microscopy (Figure 4c). As expected, the differentially regulated proteins in LDH^−/−^ cells are associated with cell death, energy production, cell growth, proliferation, differentiation, protein transport/assembly, cellular assembly, cell cycle and cell migration (Figure 4e and Appendix A).

### 3.3. Inhibition of LDHA/B significantly Increases Radiosensitivity in Tumor Cells by Impairing Stress Proteins

Hsp70 overexpression in the cytosol and on the plasma membrane of tumor cells mediates radioresistance [19,33]. LDH inhibition by Oxa and a *LDHA/B* double knockout significantly increases the radiosensitivity of LS174T WT cells (Figure 5a) and that of LS174T LDH^−/−^ and B16F10 LDH^−/−^ cells (Figure 5b,c), as exemplified by a decrease in the D_50_ value and a sensitizing enhancement ratio (SER) of more than 1.20 (Appendix A). To explain the potential mechanisms involved in the LDH-mediated radiosensitization, the membrane Hsp70 levels and cell cycle distribution were measured after irradiation with 0 (sham), 2, 4 and 6 Gy. With the increasing irradiation doses, the membrane Hsp70 expression increased in LS174T WT but remained unaltered low in LS174T LDH^−/−^ cells (Figure 5d), whereas the intracellular Hsp70 levels remained unaffected 24 h after irradiation in both WT and LDH^−/−^ cells (Figure 5e). Concomitant with an increase in radiosensitivity, LS174T LDH^−/−^ cells compared to WT cells showed a significantly increased arrest in the radiation-sensitive G2/M phase upon a single irradiation with 6 Gy (Figure 5f).

## 4. Discussion

This study demonstrates that suppressing the key oncometabolite lactate affects not only tumor cell proliferation and ROS production but also impairs the synthesis of pro-survival stress proteins, including Hsp90, Hsp70 and Hsp27. The pyruvate analog Oxa, the LDHA/B/C inhibitor GNE, glucose deprivation and a *LDHA/B* double knockout inhibit LDH activity and thereby impair the radiosensitivity of cancer cells by an interference with the stress response. The results suggest that LDHA/B may serve as an attractive target for improving the clinical outcome of radiotherapy by avoiding radioresistance via a downregulation of cytosolic and membrane-bound HSPs.

The overexpression of HSPs in a large variety of highly aggressive cancer cell types promotes cell proliferation, invasion and metastatic spread and protects tumor cells from the lethal damage induced by multiple stress factors interfering with the apoptotic pathways [34]. The expression of Hsp70 and Hsp27 is induced by the transcription factor HSF1, which also regulates glucose metabolism [35] and has been shown to increase the LDHA expression in breast cancer cells by binding to the LDHA promoter [36]. The initial findings suggest a connection between the Warburg effect and the stress response mediated by different major HSPs. Inhibiting LDH using the pyruvate analog Oxa has been shown to decrease the HSP levels in a hepatocellular carcinoma cell line [28]. Our study confirmed these findings by showing a significant decrease in the Hsp27, Hsp70 and Hsp90 levels upon the treatment of a human colorectal adenocarcinoma and a mouse melanoma cell line with a nonlethal concentration of Oxa (Figure 1b). Since, in clinical practice, the pyruvate analog Oxa has to be given continuously at very high concentrations, which would cause unfavorable side effects [37], a novel and more potent LDH inhibitor termed GNE 140 (GNE) [27] has been developed. The fact that GNE shows comparable results with respect to the inhibition of the LDH activity and HSP expression at a 6000-fold lower dose (60-mM Oxa versus 10-µM GNE) (Figure 1b) in vitro makes GNE a promising drug candidate for future applications. However, due to its rapid in vivo clearance, GNE is presently not able to sustain a LDH inhibition longer than 1 h [27]. Therefore, more stable and potent LDH/lactate inhibitors with a favorable in vivo tolerability, biodistribution and bioavailability profile are presently being investigated.

To exclude the possibility that the LDH inhibitors Oxa and GNE might directly impair the expression of HSPs or indirectly affect the stability of HSPs via yet undefined mediators, irrespective of the LDHA/B activity, tumor cells with an *LDHA/B* double knockout were assessed with respect to their effects on the HSP expression. We could demonstrate that, similar to the LDH inhibitors, a *LDHA/B* double knockout significantly reduced the intracellular HSP levels in different tumor cell types (Figure 3a,b). Moreover, LDH depletion leads to a reduced metabolic and proliferative activity (Figure 3c), which, in turn, results in a reduced ROS production (Figure 3d). LDH^−/−^ cells shift their metabolism from glycolysis to oxidative phosphorylation (OXPHOS) [20], and a decreased oxidative metabolism is often accompanied by a reduced ROS production. Our findings, which are in line with other studies, indicate that WT cells with an enhanced glycolysis exhibit higher ROS levels than the corresponding LDH^−/−^ cells [38]. One potential explanation might be that lactate, as well as pyruvate, promote ROS production by triggering mitochondrial activity, an effect which has been reported in *Caenorhabditis elegans* [39]. Another possibility is that an excess of NADH, which is essential to convert pyruvate to lactate, could stimulate electron leakage and ROS production by the respiratory complex I [40].

The heat shock response is a key survival mechanism to protect cells from lethal damage induced by a large variety of external physical and chemical stress factors, including heat, irradiation and oxygen radicals [41]. A constitutively upregulated HSP expression renders tumor cells more resistant to environmental stress and promotes therapeutic resistance [42] by not only affecting apoptosis pathways via HSPs but, also, by stabilizing core proteins of the DNA repair mechanism, such as nonhomologous end joining after double-stranded breaks, which are induced by ionizing radiation [43]. Therefore, inhibiting the cellular stress response provides a promising strategy to make tumor cells more vulnerable to anticancer therapies, including radiotherapy. In recent years, different Hsp90 and Hsp70 inhibitors have been tested in preclinical and clinical studies [44]. The Hsp90 inhibitor AUY-NVP922 showed positive effects in vitro and in vivo by increasing the radiosensitivity [45,46]. However, the efficacy of Hsp90 inhibitors is limited by their hepatotoxicity, poor water solubility, unfavorable biodistribution [44] and their compensatory upregulation of members of the antiapoptotic HSP70 family. We have been able to show that a simultaneous knockdown of HSF1, which reduces the expression of Hsp70 and Hsp27, and an inhibition of Hsp90 by AUY-NVP922 improves the radiosensitivity in human H1339 small cell lung cancer cells in vitro [47]. Due to the redundancy of the pro-tumorigenic HSP system, there is a high medical need for a universal inhibitor with a favorable toxicity profile that covers a broad spectrum of different HSPs. The findings presented herein suggest that an interference of the lactate metabolism, either by inhibiting LDH activity or a *LDHA/B* double knockout concomitantly, decrease the intracellular levels of HSF1, Hsp70, Hsp27 and Hsp90 (Figure 1b, Figure 3a,b and Appendix A). Moreover, a *LDHA/B* double knockout significantly reduces the membrane expression of Hsp70 (Figure 4a,b), which has been shown to mediate radioresistance [33] by a reduction of the Hsp70-anchoring lipid compound Gb3 in the plasma membrane and elevated extracellular Hsp70 levels. An impaired lipid metabolism in LDH^−/−^ cells has been confirmed by a comparative global proteomic analysis of the WT and LDH^−/−^ cells. Tumor cells exhibiting decreased HSP levels grow slower (Figure 3c), produce less ROS (Figure 3d) and are significantly more sensitive to radiotherapy (Figure 5a–c).

The metabolic reprogramming in tumor cells by oncogenes, like RAS and, in particular, the mutant KRAS, supports the uncontrolled tumor growth, promotes the Warburg effect and affects the glutamine/lipid metabolism, thereby mediating the therapy resistance [48]. Furthermore, an upregulation of stress granules protects tumor cells against further stress stimuli, including chemotherapeutic agents in mutant KRAS tumor cells [49]. Based on our findings, it is assumed that an inhibition of the LDH activity/Warburg effect might also impair the production of stress granules and might help to break the chemoresistance.

An increased LDH activity not only supports the HSP expression but also leads to an accumulation of lactate and acidification of the tumor microenvironment, which promotes tumor growth [18], suppresses effector T and NK cell functions [50,51,52] and supports immunosuppressive cells, including regulatory T (Treg) cells [53]. Recently, Watson et al. showed that Treg cells are flexible in their metabolism and, therefore, use lactic acid as an alternative fuel to maintain and increase their immunosuppressive function [54]. Due to the fact that high lactate levels [18] and high HSP levels can increase the radioresistance of tumor cells [19], and LDH inhibition affects both the lactate and HSP system, LDH inhibition might provide a promising strategy to improve the clinical outcome of patients with highly aggressive, therapy-resistant tumors.

## 5. Conclusions

In summary, our results indicate that an inhibition of LDH significantly increases the radiosensitivity in tumor cells by impairing the expression of several HSPs with antiapoptotic capacities. Therefore, in the future, a pharmacological inhibition of LDH might provide a promising strategy to increase the effectiveness of radiotherapy. Combined therapeutic approaches, including a pharmacological targeting of LDH during radiotherapy, have the capacity to improve tumor killing with lower radiation doses whilst sparing normal tissues, thereby reducing the normal tissue toxicities.

## Figures and Tables

**Figure 1 cancers-13-03762-f001:**
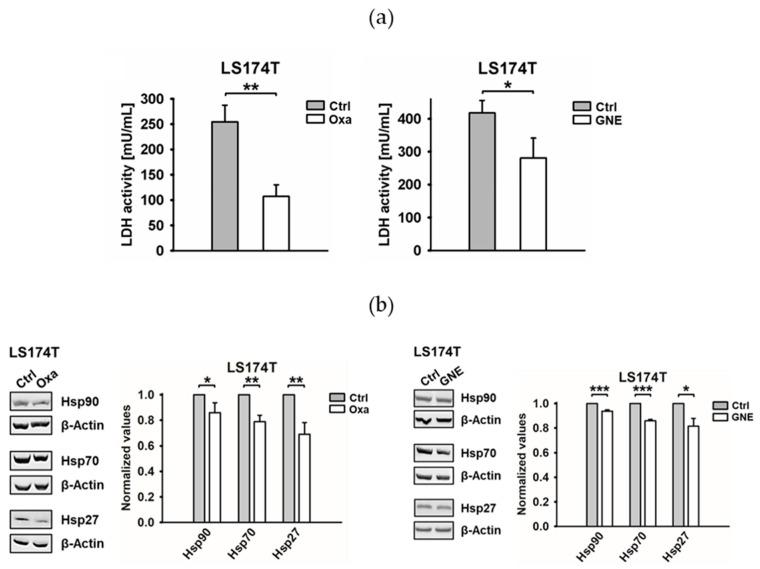
Lactate dehydrogenase (LDH) inhibition by oxamate (Oxa) and GNE-140 (GNE) decreases the expression of Hsp90, Hsp70 and Hsp27. (**a**) The effect of Oxa (60 mM, 48 h) and GNE (10 µM, 24 h) on LDH activity in LS174T cells (*: *p* ≤ 0.05 and **: *p* ≤ 0.01). (**b**) Representative immunoblot showing the intracellular expression of Hsp90, Hsp70 and Hsp27 in untreated (Ctrl) and Oxa- or GNE-treated LS174T cells. β-Actin was used as a loading control. The quantification of the heat shock protein (HSP) expression levels is shown in the adjacent bar chart. Error bars show the standard deviations (SD) of at least three biological replicates (*: *p* ≤ 0.05, **: *p* ≤ 0.01 and ***: *p* ≤ 0.001). Full Western blot images are available in Appendix A.

**Figure 2 cancers-13-03762-f002:**
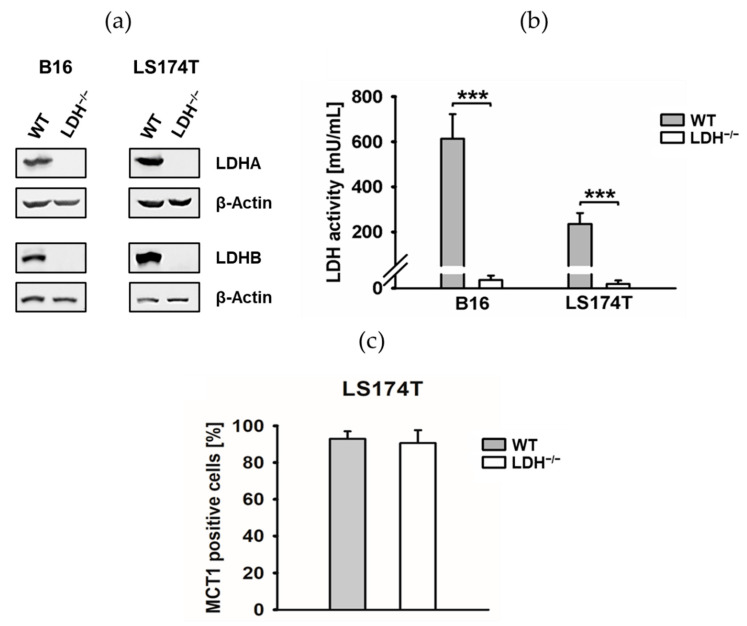
*LDHA/B* double knockout and its effect on lactate dehydrogenase (LDH) activity. (**a**) Representative immunoblot showing the successful double knockout of *LDHA* and *LDHB* (referred to as LDH^−/−^) of mouse B16F10 and human LS174T tumor cell lines. β-Actin was used as a loading control. (**b**) The effect of *LDHA/B* double knockout on LDH activity in B16F10 and LS174T cells (***: *p* ≤ 0.001). (**c**) Flow cytometric analysis of the lactate transporter MCT1 using MCT1-APC mAb on LS174T WT and LDH^−/−^ cells. The proportion of positively stained cells is shown. Error bars show the standard deviations of at least three biological replicates. Full Western blot images are available in Appendix A.

**Figure 3 cancers-13-03762-f003:**
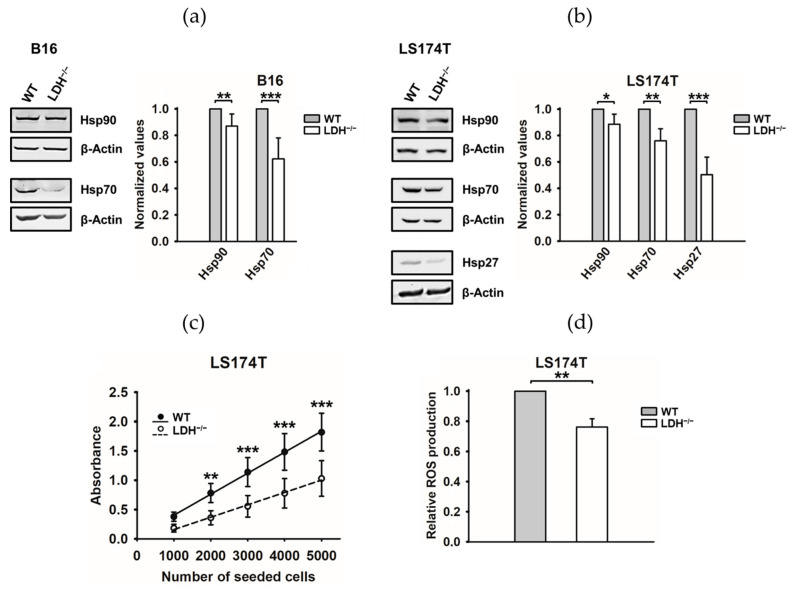
*LDHA/B* double knockout (LDH^−/−^) inhibits the expression of heat shock proteins (HSP) Hsp90, Hsp70 and Hsp27 and inhibits the proliferative activity and reactive oxygen species (ROS) production in tumor cells. (**a**,**b**) Representative immunoblot showing the expression of intracellular Hsp90, Hsp70 and Hsp27 of B16F10 (**a**) and LS174T (**b**) cells. Quantification of the HSP expression levels are shown in the adjacent bar chart. Error bars show the standard deviations (SD) of at least four biological replicates (*: *p* ≤ 0.05, **: *p* ≤ 0.01 and ***: *p* ≤ 0.001). (**c**) A cell proliferation assay (CCK-8) demonstrated that *LDHA/B* double knockout inhibited the proliferation in LS174T cells (**: *p* ≤ 0.01 and ***: *p* ≤ 0.001). (**d**) The DCFDA assay was used to determine the intracellular ROS levels in LS174T cells, with fluorescence being measured using a microplate reader. Error bars show the SD of four biological replicates (**: *p* ≤ 0.01). Full Western blot images are available in Appendix A.

**Figure 4 cancers-13-03762-f004:**
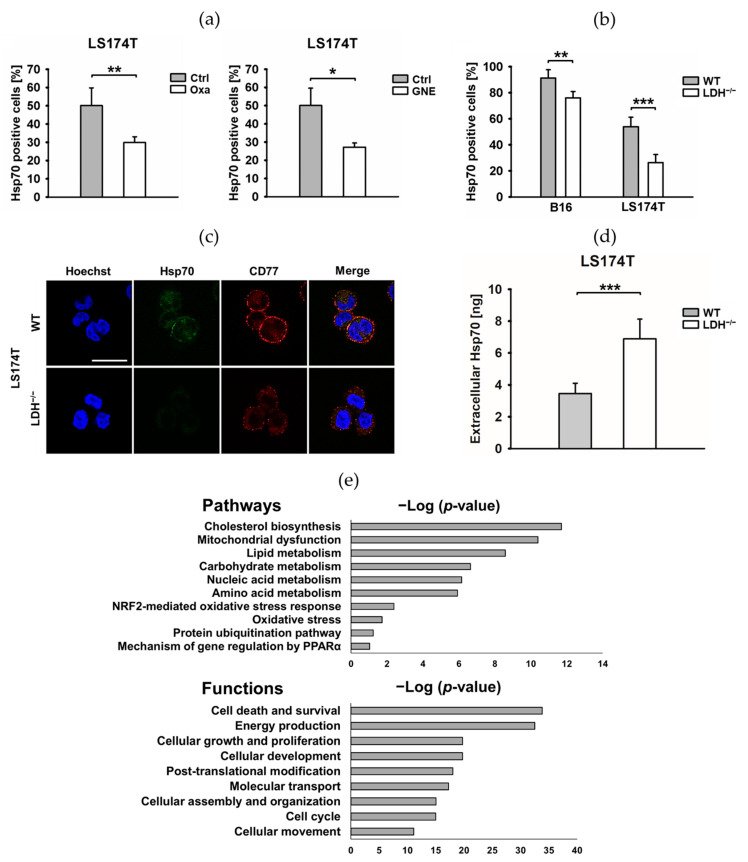
The lactate dehydrogenase (LDH) inhibition by oxamate (Oxa) and GNE-140 (GNE) and a *LDHA/B* double knockout (LDH^−/−^) results in a reduced heat shock protein 70 (Hsp70) and globotriaosylceramide (Gb3) expression on the cell membrane. (**a**,**b**) Membrane-bound Hsp70 on untreated (Ctrl) and Oxa- or GNE-treated LS174T tumor cells (**a**) and membrane-bound Hsp70 on B16F10 and LS174T cells (**b**), as determined by flow cytometry using cmHsp70.1-FITC mAb. The proportion of positively stained cells is shown. Error bars show the standard deviations (SD) of at least three biological replicates (*: *p* ≤ 0.05, **: *p* ≤ 0.01 and ***: *p* ≤ 0.001). (**c**) LS174T cells stained for the expression of membrane Hsp70 (cmHsp70.1-FITC mAb, green) and Gb3 (CD77-PE mAb, red) were analyzed by confocal microscopy. The colocalization of Hsp70 and Gb3 is visualized in yellow as a merge of the red and green staining. Scale bar: 5 µm. (**d**) Extracellular Hsp70 levels in the supernatant of the LS174T cells were measured by ELISA and the data normalized to 1 × 10^6^ viable tumor cells. Error bars show the SD of five biological replicates (***: *p* ≤ 0.001). (**e**) Proteome profiling of LS174T WT and LDH^−/−^ cells. The most significant canonical pathways and top molecular functions altered by LDH depletion are illustrated. The analyses were generated through the use of Ingenuity Pathway Analysis (IPA) (QIAGEN Inc., https://www.qiagenbio-informatics.com/products/ingenuity-pathway-analysis, accessed on 22 April 2021). Bars indicate the canonical pathways, and the *y*-axis displays the −(log *p*) enrichment significance. NRF2: nuclear factor erythroid 2–related factor 2, PPARα: peroxisome proliferator-activated receptor alpha.

**Figure 5 cancers-13-03762-f005:**
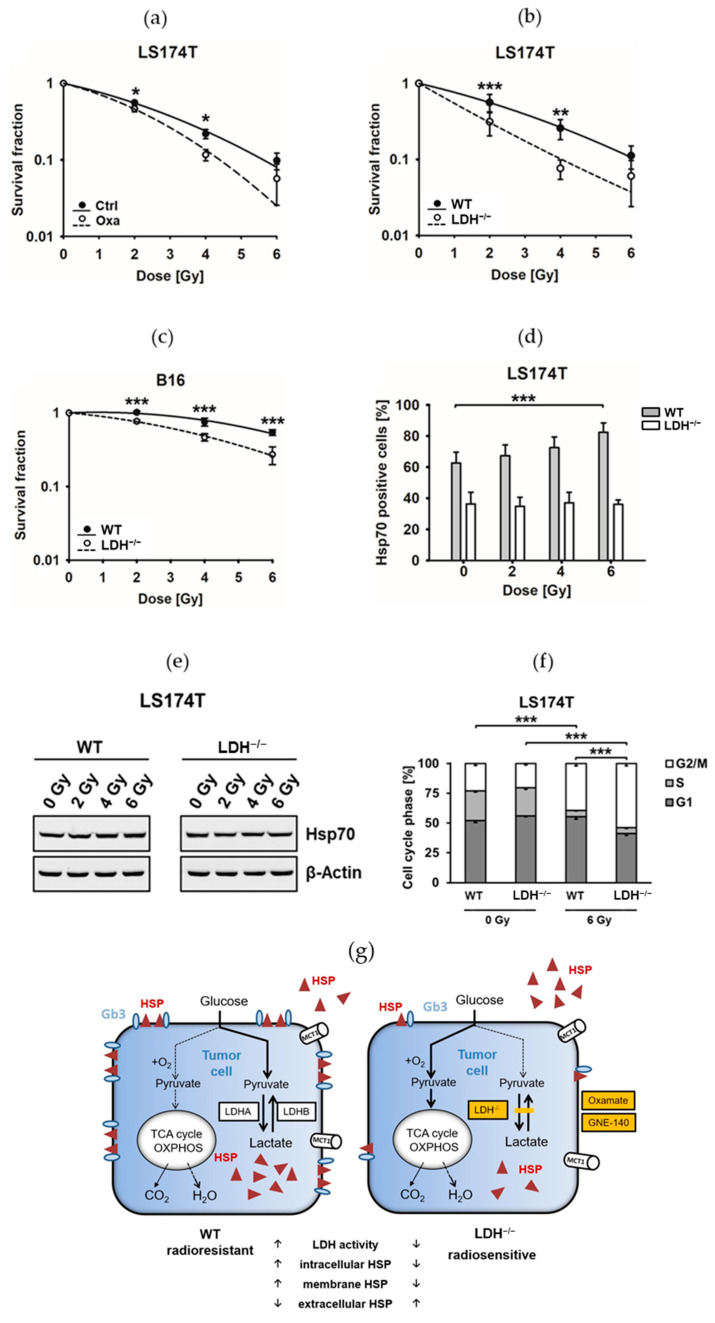
Inhibition of the LDHA/B activity significantly increases radiosensitivity in LS174T and B16F10 cells. (**a**) Colony-forming assay of untreated (Ctrl) and Oxa-treated LS174T cells after irradiation with 0 Gray (Gy, sham), 2 Gy, 4 Gy and 6 Gy (*: *p* ≤ 0.05). (**b**,**c**) Colony-forming assay of LS174T (**b**) and B16F10 (**c**) and wildtype (WT) and *LDHA/B* double knockout LDH^−/−^ cells after irradiation with 0 Gy (sham), 2 Gy, 4 Gy and 6 Gy (**: *p* ≤ 0.01 and ***: *p* ≤ 0.001). (**d**) Percentage of LS174T WT and LDH^−/−^ cells expressing membrane heat shock factor 70 (Hsp70) 24 h after irradiation with 0 Gy (sham), 2 Gy, 4 Gy and 6 Gy. Bars represent the mean value and the corresponding standard deviations (SD) of four independent experiments (***: *p* ≤ 0.001). (**e**) Representative immunoblot showing cytosolic Hsp70 expression by LS174T WT and LDH^−/−^ cells 24 h after irradiation with 0 Gy (sham), 2 Gy, 4 Gy and 6 Gy. β-Actin was used as a loading control. (**f**) LS174T WT and LDH^−/−^ cells were irradiated with 6 Gy. Twenty-four hours after irradiation, cells were fixed, and the cell cycle distribution was determined by flow cytometry. The mean value and the corresponding SD of three independent experiments is shown (***: *p* ≤ 0.001). (**g**) Schematic illustration of the major findings: An impaired LDH activity induced by an *LDHA/B* double knock out (LDH^−/−^) or by the LDH inhibitors (Oxa and GNE) increases the radiosensitivity of tumor cells by a generalized downregulation of cytosolic pro-survival stress proteins, a reduction in plasma membrane-bound Hsp70 and reduced lactate levels-although the lactate transporter MCT1 remains unaffected-and an impaired lipid metabolism. Elevated extracellular Hsp70 levels are associated with a reduction in Hsp70 membrane-anchoring glycosphingolipid Gb3. Full Western blot images are available in Appendix A.

## Data Availability

The mass spectrometry proteomics data were deposited to the Proteo-meXchange Consortium via the PRIDE partner repository with the dataset identifier PXD026078.

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
