# Peer review of "Targeting Cancer Metabolism Breaks Radioresistance by Impairing the Stress Response"

_cancers, 2021, doi:10.3390/cancers13153762_

Round 1

Reviewer 1 Report

Article by Dr. Multhoff and group exploring the possibility of targeting metabolism to combat radioresistance and its involvement in the impairment of stress response. It's a novel study in a timely manner. But few things need to be addressed before it is ready for acceptance, they are as follows:

  1. It has been shown recently that ( 10.1158/0008-5472.CAN-19-1363) targeting glutamine metabolism can be efficacious for combatiing chemoresistance. Authors should mention this in their introduction part. It will be relevent to add.
  2. Also, it has been discussed in (doi: 10.1038/s43018-021-00184-x) the global role of metabolism in combating resistance in RAS-driven cancers, few lines needs to added on this context in discussion part. It will helpful for readers to understand the broader aspect of current manuscript.
  3. Authors should add a model in last figure depicting the overall message of this manuscript. 
  4. Another aspect is whether this study plays any role in stress granules. This will be one of the future aspects of this study. They relevant references can be found in doi: 10.1038/s43018-021-00184-x and https://doi.org/10.1016/j.cell.2016.11.035

Author Response

Point-by-point letter

Reviewer 1:

Article by Dr. Multhoff and group exploring the possibility of targeting metabolism to combat radioresistance and its involvement in the impairment of stress response. It's a novel study in a timely manner. But few things need to be addressed before it is ready for acceptance, they are as follows:

  1. It has been shown recently that (DOI: 10.1158/0008-5472.CAN-19-1363) targeting glutamine metabolism can be efficacious for combating chemoresistance. Authors should mention this in their introduction part. It will be relevent to add.

Answer: This aspect has been addressed and the relevant reference was included into the introduction.

  1. Also, it has been discussed in (doi: 10.1038/s43018-021-00184-x) the global role of metabolism in combating resistance in RAS-driven cancers, few lines needs to added on this context in discussion part. It will helpful for readers to understand the broader aspect of current manuscript.

Answer: The reference was included and discussed as recommended.

  1. Authors should add a model in last figure depicting the overall message of this manuscript. 

Answer: This point is well taken and a model was included as a new Figure.

  1. Another aspect is whether this study plays any role in stress granules. This will be one of the future aspects of this study. They relevant references can be found in doi: 10.1038/s43018-021-00184-x and https://doi.org/10.1016/j.cell.2016.11.035

Answer: This aspect has been addressed in the discussion part.

Reviewer 2:

This is well written paper connecting cancer metabolism and radioresistance.

The authors contribute the following major conclusion from their experimental approach, which is important to understand the radioresistance and sensitivity of cancer cells.

  1. The connection between metabolic activity (lactate) increases in ROS, and HSP expression results in radioresistance and proliferation is well documented. 
  2. The absence of LDH affects the membrane-bound HSP70, and the authors show that is due to the reduction of GB3. A possible explanation is given by proteomic studies, suggesting deregulated lipid metabolism. 
  3. The absence of LDH leads to reduced HSP, which causes the radiosensitivity. 

The following should be included in the discussion:

There is no mention of DNA repair mechanism, specifically Double-strand break repair which results from radiation and the connection between HSP.  

Answer: This aspect has been addressed. A novel reference has been included.

References are appropriate.

The authors want to thank all reviewers and the editor for constructive and helpful comments.

Reviewer 2 Report

This is well written paper connecting cancer metabolism and radioresistance.

The authors contribute the following major conclusion from their experimental approach, which is important to understand the radioresistance and sensitivity of cancer cells.

  1. The connection between metabolic activity (lactate) increases in ROS, and HSP expression results in radioresistance and proliferation is well documented. 
  2. The absence of LDH affects the membrane-bound HSP70, and the authors show that is due to the reduction of GB3. A possible explanation is given by proteomic studies, suggesting deregulated lipid metabolism. 
  3. The absence of LDH leads to reduced HSP, which causes the radiosensitivity. 

The following should be included in the discussion:

There is no mention of DNA repair mechanism, specifically Double-strand break repair which results from radiation and the connection between HSP.  

References are appropriate.

Author Response

(The authors gave the same response as above.)

Round 2

Reviewer 1 Report

All concerns have been addressed. Ready for acceptance.